# Allelic Diversification of the *Wx* and *ALK* Loci in Indica Restorer Lines and Their Utilisation in Hybrid Rice Breeding in China over the Last 50 Years

**DOI:** 10.3390/ijms23115941

**Published:** 2022-05-25

**Authors:** Li-Xu Pan, Zhi-Zhong Sun, Chang-Quan Zhang, Bu Li, Qing-Qing Yang, Fei Chen, Xiao-Lei Fan, Dong-Sheng Zhao, Qi-Ming Lv, Ding-Yang Yuan, Qiao-Quan Liu

**Affiliations:** 1Jiangsu Key Laboratory of Crop Genomics and Molecular Breeding, State Key Laboratory of Hybrid Rice, Jiangsu Key Laboratory of Crop Genetics and Physiology, Yangzhou University, Yangzhou 225009, China; panlixu288@163.com (L.-X.P.); cqzhang@yzu.edu.cn (C.-Q.Z.); libu1026@126.com (B.L.); y19895320793@163.com (Q.-Q.Y.); feich12345@outlook.com (F.C.); xlfan@yzu.edu.cn (X.-L.F.); dszhao@yzu.edu.cn (D.-S.Z.); 2State Key Laboratory of Hybrid Rice, Hunan Hybrid Rice Research Center, Changsha 410125, China; szznihaoa@163.com (Z.-Z.S.); lvqiming00@163.com (Q.-M.L.); 3Key Laboratory of Plant Functional Genomics of the Ministry of Education, Jiangsu Co-Innovation Center for Modern Production Technology of Grain Crops, College of Agriculture, Yangzhou University, Yangzhou 225009, China

**Keywords:** *Oryza sativa* L., hybrid rice, ECQ, *Waxy*, *ALK*, *indica* restorer, rice breeding

## Abstract

Hybrid rice technology has been used for more than 50 years, and eating and cooking quality (ECQ) has been a major focus throughout this period. *Waxy* (*Wx*) and alkaline denaturation (*ALK*) genes have received attention owing to their pivotal roles in determining rice characteristics. However, despite significant effort, the ECQ of restorer lines (RLs) has changed very little. By contrast, obvious changes have been seen in inbred rice varieties (IRVs), and the ECQ of IRVs is influenced by *Wx*, which reduces the proportion of *Wx^a^* and increases the proportion of *Wx^b^*, leading to a decrease in amylose content (AC) and an increase in ECQ. Meanwhile, *ALK* is not selected in the same way. We investigated *Wx* alleles and AC values of sterile lines of female parents with the main mating combinations in widely used areas. The results show that almost all sterile lines were *Wx^a^*-type with a high AC, which may explain the low ECQ of hybrid rice. Analysis of hybrid rice varieties and RLs in the last 5 years revealed serious homogenisation among hybrid rice varieties.

## 1. Introduction

Increasing rice (*Oryza sativa* L.) yield has been the main breeding objective for several decades, and the yield potential of irrigated rice has already experienced two quantum leaps [1]. The first advance was semidwarf breeding [2,3], and the second was utilisation of hybrid rice technology [4]. In 1971, the team of Professor Yuan Longping discovered *O. rufipogon*, a wild male sterile plant, in Hainan Province, China, which revealed the production potential of hybrid rice, paving the way for the development of F_1_ hybrid rice introduced in 1976 [4]. Through the introduction of hybrid rice, the development of heterotic markers led to the second breakthrough and consequent leap in production. The achievements of the 1960s and 1970s increased rice production by ~50% in 10 years in many countries, including China, Indonesia and Vietnam. Since then, hybridisation has been the core focus of rice breeding in China [5,6].

Although *xian/indica*, *XI* (*O. sativa* L. *subsp. xian/indica*)-*geng/japonica*, *GJ* (*O. sativa* L. *subsp. geng/japonica*) and *GJ-GJ* hybrids have been reported, the most common rice hybrids are *XI-XI* hybrids, or *XI* hybrids for short. There are two types of *XI* hybrid rice: three-line hybrids and two-line hybrids [7,8,9,10,11]. In the early stages, there were many materials collected from various countries, and then more and more materials began to be cultivated in China. Among them, South China (SC) took the lead in breeding a large number of varieties. Since the 1990s, a large number of materials have been bred in the upper reaches of the Yangtze River (UYR) and the middle and lower reaches of the Yangtze River (MLYR). Among the materials in the UYR, 3-RLs predominate, while 2-RLs are mainly from the MLYR. Hybrid rice ECQ is directly controlled by both restorer and sterile lines.

As people’s living standards increase with economic development, rice ECQ receives attention. In hybrid rice breeding, a high and stable yield is the main goal, and rice ECQ indices are essential to this end [12,13]. Primary factors affecting rice ECQ are AC, gelatinisation temperature (GT), gel consistency (GC) and viscosity [14,15,16]. For a long time, it was generally believed that hybrid rice in China had a high yield but poor quality [17]. Higher AC and a chalky grain rate are the main reasons why the quality of hybrid rice is lower than that of inbred rice [18]. In order to improve the efficacy of breeding for rice quality, the Ministry of Agriculture and Rural Affairs of China promulgated the agricultural industry standards NY20-1986 (1986), ‘quality edible rice’, and NY/T-593-2002 (2002), ‘cooking rice variety quality’.

These factors are mainly regulated by starch-synthesis-related genes. Starch is synthesised by the orchestrated functional interactions of four classes of enzymes; ADP-glucose pyrophosphorylase, starch branching enzyme (BE), starch synthase (SS) and starch debranching enzyme (DBE) [19,20,21,22,23]. The granule-bound starch synthase I enzyme (GBSSI) is required for amylose synthesis in rice, and the gene encoding GBSSI was named *Waxy* (*Wx*) [24]. Many alleles of *Wx* (such as *Wx^a^*, *Wx^b^*, *Wx^in^*, *Wx^op^*, *Wx^mp^*, *Wx^lv^, wx* and *Wx^mw/la^*) lead to regional changes in rice AC and affect consumer preferences [24,25,26,27,28,29,30]. GT is mainly controlled by the alkaline denaturation (*ALK*) gene encoding modified starch synthase IIa (SSIIa) [31]. Several studies reported that at least three single-nucleotide polymorphisms (SNPs) of *ALK* are associated with the diversity of GT in rice. Ex8-733 bp (A/G) and Ex8-864/865 bp (G/T and C/T) of *ALK* in exon 8 (Ex8) are closely related to changes in GT [14,31,32,33]. These functional SNPs produce three haplotypes/alleles, including *ALK^a^* (A-GC) and *ALK^b^* (G-TT), which control low GT, and *ALK^c^* (G-GC), which controls high GT [32,34]. Recently, a new *ALK* allele, *ALK^d^* Ex1-294 (G/T), was reported [15].

For rice, seed development begins with double fertilisation, which leads to the development of the embryo and the endosperm. After double fertilisation, two sperm enter the embryo sac through the pollen tube. One sperm cell fuses with the egg cell to form a zygote, and the other sperm cell fuses with the central cell to form a triploid primary endosperm cell. Zygotes and primary endosperm cells develop into an embryo and transfer genetic material from parents to the next generation, and endosperm nourishes developing embryos/seedlings [35,36,37]. Both sterile lines and restorer lines will affect the ECQ of hybrid rice. Although ECQ evolutionary trends in hybrid rice breeding can be realised directly from the perspective of restorer lines, the process of improving the ECQ of hybrid rice can be achieved indirectly.

Various materials are employed for the cultivation of parents of hybrid rice, especially RLs, which have been mainly used to systematically explore the ECQ of hybrid rice. At present, there are few studies on the ECQ of hybrid rice. Previous studies have mainly focused on rice quality in the F_2_ generation of hybrid rice [13,38,39]. However, the results are limited to specific combinations and cannot be widely used to guide the cultivation of hybrid rice. Therefore, based on the analysis of restorer line materials, we investigated improving RLs to indirectly predict the evolutionary process driving the ECQ of hybrid rice.

## 2. Results

### 2.1. Characteristics of ECQ among Indica RLs

RL materials were subjected to various analyses, including AAC, differential scanning calorimetry (DSC), Rapid Visco-Analyzer (RVA) and ECQ (Figure 1c–h and Appendix A). AAC values fell into two main ranges: 10–18% and 22–30%. Similarly, *T_p_* values fell into two main ranges: 65–75 °C and 75–85 °C. The results reveal a negative correlation between AAC and taste value (r = −0.515, *p* < 0.01; Appendix A). *T_p_* was not significantly correlated with AAC or taste value, but *T_p_* was significantly correlated with BDV and SBV, positively in the case of BDV (r = 0.517, *p* < 0.01; Appendix A) and negatively in the case of SBV (r = −0.511, *p* < 0.01; Appendix A).

### 2.2. ECQ Differentiation with Years and Places

In the past 50 years, none of the indices of RL materials have changed significantly. By contrast, for IRVs, AAC dropped from ~25% in 1970 to ~15% in 2000 (stage 1), and AAC has remained stable at ~15% since (stage 2). Taste value increased over time before the 21st century, and has remained stable since. *T_p_* was high before 2000, and has decreased since. RVA and other indices followed similar trends. There was no significant change in RLs over time, but there was an obvious regular pattern for IRVs (Figure 1c–h and Appendix A). Regarding rice ECQ, this gradually improved over time for IRVs, but remained stable for RLs, although it remained higher for RLs than for IRVs.

According to the classification of the area used for collecting materials, all materials were divided into four groups: Foreign, SC, UYR and MLYR. Since entries for the Foreign group were fewer and less distributed than in other areas after introduction, this group did not receive our focus. In general, *T_p_* and ECQ values for UYR were higher than for other regions. Differences in hardness, stickiness and PKV were identical in the overall trend compared to ECQ (Appendix A). Regarding hardness and SBV, 2-RLs were significantly higher than 3-RLs, and PKV and BDV displayed the opposite phenomena. Except for these differences in indicators, there were no significant differences between 2-RLs and 3-RLs. All ECQ indices were significantly different between IRVs and RLs. In general, IRVs had higher AAC values and lower GT and ECQ values than RLs (Appendix A).

### 2.3. Allelic Diversification of the Wx Locus in Indica RLs

The data used in this experiment were obtained from the resequencing data of 1143 xian/indica rice lines reported by Lv et al., [11]. *Wx* and *ALK* segments were extracted from the data (physical location: Shuhui 498, R498). R498 genome assembly and annotation can be found at http://www.mbkbase.org/R498 (accessed 8 August 2021). Analysis of the *Wx* haplotype was performed using the chromosome 6 physical interval (1643113–1648065). A total of 33 SNPs were found to be in linkage disequilibrium with index SNPs (Figure 2b). Five haplotypes were identified from the above SNPs, and AAC values for materials corresponding to these haplotypes showed differences, but there were no significant differences between haplotypes 4 and 5 (Figure 2d). According to the analysis of known functional loci, there were no differences between haplotypes 4 and haplotype 5, which indicates that (508) in the intron and (215) in exon C-T changed CCC (proline) to CCT (proline); hence, there was no change in amino acids due to codon degeneracy.

Four *Wx* alleles were detected in the RLs. The predominant *Wx* gene was *Wx^b^* (228), followed by *Wx^a^* (44), and *Wx^lv^* and *Wx^in^* were also detected in smaller numbers (Figure 2a). The Hap1 material was *Wx^lv^*-type, and Hap1 and *Wx^lv^* were consistent in functional sites. The AAC of Hap1 was >25%, consistent with *Wx^lv^*-type rice. Similarly, Hap2 material was *Wx^a^*, whereas Hap3 material was *Wx^in^*. It is worth noting that Hap4 and Hap5 were *Wx^b^*, and these were consistent at functional sites. The AAC of Hap4 and Hap5 was ~15%, consistent with *Wx^b^*-type rice (Figure 2d,e). Additionally, the *Wx* genes of IRV materials were also analysed (Appendix A).

Based on comprehensive analysis of breeding years, materials were divided into seven time periods: <1980, 1980–1989, 1990–1999, 2000–2004, 2005–2009, 2010–2014 and ≥2015 (Figure 3a,d,g). The AAC of *Wx^a^* was ~25%, and for *Wx^b^*, it was ~15%, and the higher the proportion of *Wx^a^*, the greater the increase in AAC. According to the combined genotype and phenotype data, the reason for the decrease in AAC for IRVs before the 2000s was that the allelic proportion of the *Wx^b^* type increased, and the proportion remained stable (~15%) after the 2000s (Appendix A). The allelic proportion of the *Wx^b^* type of 2-RLs also increased over time, but this was masked by 3-RLs. It should be noted that the early stage of 2-RLs was screened by test crossing with IRVs. The proportion of the *Wx^b^* allele did not differ between areas (Appendix A). Evidently, the *Wx^b^*-type allele of IRVs was scarcer than that of RLs. Analysis of different types of materials showed that the *Wx^a^*/*Wx^b^* ratio of IRVs and 2-RLs decreased, while the ratio of 3-RLs did not change significantly (Figure 3a,d,g and Appendix A).

### 2.4. Allelic Diversification of the ALK Locus in Indica RLs

Analysis of *ALK* haplotype was performed using chromosome 6 physical interval 6811279–6816183. A total of 30 SNPs were found to be in linkage disequilibrium with index SNPs (Figure 4b). Five haplotypes were identified by the SNP above, and the *T_p_* values of materials corresponding to these haplotypes showed differences, but it should be noted that the *T_p_* of haplotype 1 and another four haplotypes were significantly different (Figure 4c,d). According to known functional loci analysis, there were differences between haplotypes 1 and the other four haplotypes, but there were no differences between haplotypes 2–5. In addition to 6811957, 6812194, 6815238, 6815273, 6815342 and 6815357, other SNP differences were found in the noncoding region, and except for 6815238 (A–G) encoding a serine to glycine change, all amino acids remained the same. According to known functional loci analysis, there were no differences between haplotypes 2 and 5; there were only differences between haplotype 1 and other haplotypes in functional loci of Ex8-864/865 (Figure 3d).

Additionally, the *ALK* genes of IRV materials were analysed (Appendix A). Two *ALK* alleles were detected in RLs (Figure 3a), most of which were *ALK^b^* (134), followed by *ALK^c^* (141). Haplotypes 2–5 were different in SNPs, but the GT was similar with no significant differences between haplotypes. Relative differences between haplotypes were further explored by cluster analysis of *ALK* genes with rice varieties as controls. A phylogenetic tree and genetic distance analyses showed that all varieties could be divided into three categories: Hap1 and 4, Hap2 and 3 and Hap5. The Hap1 allele clustered with the Hap4 allele, indicating a close genetic relationship between these two alleles, and they were distributed in *XI* and *GJ*. Hap2, Hap3 and Hap5 were distributed in *XI*, and Hap2 and Hap3 were closely genetically related (Appendix A).

Previous studies showed that *ALK* had three major alleles: *ALK^a^*, *ALK^b^* and *ALK^c^*. There were only two types (*ALK^b^* and *ALK^c^*) in the materials studied herein. Hap1 materials were *ALK^b^*, and Hap1 and *ALK^b^* were consistent at functional sites. The *T_p_* of Hap1 was ~70 °C, consistent with the *T_p_* of the *ALK^b^* type. Hap2–5 materials were *ALK^c^*, and they were consistent in functional sites. The *T_p_* of Hap2–5 was ~80 °C, consistent with the *T_p_* of the *ALK^c^* type (Figure 3d,e).

The *ALK* gene was divided into *ALK^b^* and *ALK^c^* types, and *ALK^c^* had a higher GT than *ALK^b^*. As a result, the *ALK^b^*-type rate of 2-RLs and 3-RLs did not change regularly over time; rather, fluctuations were irregular. The allelic proportion of the *ALK^b^*-type rate of IRVs displayed two stages. The first stage (before the year 2000) had a lower proportion (~50%) than the second stage (after 2000), which was ~95% (Figure 3b,e,h and Appendix A). Regarding area distribution, the allelic proportion of the *ALK^b^* type in the UYR was lower than in SC and the MLYR, which led to *T_p_* in the UYR being significantly higher than that in SC and the MLYR (Appendix A). The proportion of the *ALK^b^* haplotype was higher in IRVs than RLs (Figure 3b,e,h and Appendix A). For the *ALK* gene, the *ALK^b^*/*ALK^c^* ratio of RLs did not change significantly, but for IRVs, the ratio displayed an increasing trend (Appendix A).

### 2.5. Combining Wx and ALK Alleles and Their Utilisation in Indica RLs

*Wx* and *ALK* are both on the short arm of chromosome 6, with physical positions 164313−164806 and 6811279−6816183, respectively, in Nipponbare. Because *Wx* was mainly composed of *Wx^a^* and *Wx^b^*, the two genes are divided into four types: *Wx^a^*/*ALK^b^*, *Wx^a^*/*ALK^c^*, *Wx^b^*/*ALK^b^* and *Wx^b^*/*ALK^c^.* For different types of materials, we performed analysis of *ALK* and *Wx* gene combinations (Appendix A), and there were no significant differences between *ALK^b^* and *ALK^c^* in IRVs, 2-RLs or 3-RLs under the *Wx^a^* background. There were no significant differences between *ALK^b^* and *ALK^c^* in RLs under the *Wx^b^* background, but the gene frequency of *ALK^b^* was much higher than that of *ALK^c^* in IRVs. This shows that there was selective pressure on *ALK* under the *Wx^b^* background for IRVs. Previous results show that the *Wx^b^* type has been widely used in rice breeding over the years (Appendix A). Thus, it was speculated that different alleles of the *ALK* gene were not selected in the early breeding process, but began to be selected in the later stages. To explore this further, the distribution of breeding time under different *Wx* backgrounds was analysed. The results show that *Wx^a^* materials were generally bred in the early stages, while *Wx^b^* materials were generally bred more recently, consistent with the prediction.

### 2.6. Contributions of Elite Ls in the Development of Hybrid Rice

In this experiment, 275 RLs were selected, including 2-RLs and 3-RLs. Hybrid rice varieties bred using these RLs were collected and analysed. The results show that during this period, 80 RLs were used in hybrid rice breeding, and 1057 varieties were bred (https://www.ricedata.cn/variety/) (accessed 1 January 2022). Moreover, 24 RLs served as male plants in the breeding of >10 hybrid rice varieties (Figure 5a and Appendix A). According to the statistical results of the number of hybrid rice varieties bred using RLs, almost all *Wx* genotypes of the elite RLs were *Wx^b^*, while the genotypes *ALK^b^* and *ALK^c^* were almost equal among *ALK* genes (Figure 5a). This shows that from the perspective of RLs used in hybrid rice breeding, elite RLs mainly used *Wx^b^* with a low AC, while *Wx^a^* with a high AC has rarely been used. The two main alleles (*ALK^b^* and *ALK^c^*) of the *ALK* gene regulating GT are not present in RLs used in hybrid rice breeding.

In general, regarding RLs, the selection of functional alleles and the ECQ of the *Wx* gene of elite RLs used in hybrid rice breeding met the requirements of high ECQ. Notably, 132 hybrid rice varieties were bred with Huazhan as RLs, accounting for 12.49% of the total (Figure 5b). Huazhan was bred by the China National Rice Research Institute and Guangdong Academy of Agricultural Sciences. The breeding departments of these varieties were distributed all over China, and suitable planting areas were also widely distributed. Another extremely important RL in the history of hybrid rice breeding is Minghui63, used as a male plant to breed 35 hybrid rice varieties, considerably fewer varieties than Huazhan (Figure 5a). The possible reason for this is that after 2014, the state successively launched the green channel for variety approval and the consortium test channel, and methods for testing rice varieties diversified, resulting in a sharp increase in the number of rice varieties included in regional tests, and it was ultimately approved. The above results confirm that in the past 5 years of hybrid rice breeding, a few materials have been heavily relied upon; in addition to Huazhan, Wushansimiao (28) and Chenghui727 (24) have been widely applied (Figure 5c). Further details are included in Appendix A.

## 3. Discussion

Herein, we assessed 275 rice samples from RLs of foreign, SC, UYR and MLYR origin. Earlier materials were mainly introduced from foreign sources, but a large number of materials were developed in SC, followed by more in the Yangtze River region. The 3-RLs were evenly distributed in different areas, and 2-RLs were mainly distributed in MLYR.

The phenotypes of selected materials were investigated, including AAC, DSC, RVA and taste value, and the results are consistent with the ranges reported in previous studies [14,15,29,38,39]. Correlation analysis of different indices revealed a significant correlation between AAC and taste value (r = −0.515, *p* < 0.01), with a significant negative correlation. Specifically, varieties with a higher AAC had higher hardness and a lower taste value [14]. There was no significant correlation between GT and ECQ, consistent with previous studies [15]. Significant correlations were identified between certain RVA profiles and texture characteristics. Therefore, RVA profiles are commonly used to evaluate milled rice ECQ [40,41]. Our experimental results reveal a significant positive correlation between eating value and BDV, and a significant negative correlation with SBV. These results are consistent with previous reports.

In the past 50 years, indices of RL materials have not changed regularly over time. Rather, RLs introduced from foreign sources were mainly *Wx^b^*-type, and this appears to be reflected in ECQ. RLs introduced in the later stages were mainly used in recent breeding, resulting in low AAC. By contrast, obvious regular changes occurred in IRVs. Analysis of AAC of IRVs over this time interval, from the 1970s to the 2000s, showed a decreasing trend with a final decrease to ~15%. After the 2000s, AAC stabilised at 15% without any further decline. Reducing AAC could improve the ECQ of rice. There was no clear standard for ECQ in China before the first standard NY20-1986 (1986), ‘quality edible rice’, was issued in the 1980s. NY122-86, including ECQ indicators, limited the permissible range of AAC, GC and GT in high-quality rice. There is a significant negative correlation between ECQ and AAC, since reducing AAC to improve rice ECQ has become a major goal of plant breeders. By contrast, GT has no significant correlation with ECQ, and the GT of different grades of rice is identical among the same rice types of rice, reflecting the fact that GT has not been considered in the breeding process. The Ministry of Agriculture and Rural Affairs of China promulgated new agricultural industry standards in the form of NY/T 593-2013, ‘cooking rice variety quality’. This includes GT corresponding to alkali spreading value (ASV) > 6 or GT < 70 °C, which will encourage rice breeders to cultivate varieties with a lower GT.

*Wx* and *ALK* are the main genes regulating rice ECQ. *Wx* encodes the soluble starch synthase GBSS1, and many allele variations of *Wx* have been identified, including *Wx^a^*, *Wx^b^*, *Wx^in^*, *Wx^op^*, *Wx^mp^ Wx^lv^*, *Wx^mw^/^la^* and *wx*, leading to regional changes in AC that affect consumer preferences [24,25,26,27,28,29,42,43]. The *Wx^b^* allele type was found to be predominant in the experimental materials, followed by *Wx^a^*, while *Wx^lv^* and *Wx^in^* were also present. Over the years, the *Wx^a^*/*Wx^b^* ratio of RL materials did not change regularly; rather, *Wx^a^*/*Wx^b^* of 2-RLs and IRVs decreased from the 1970s to the 2000s, then stabilised after then 2000s, consistent with changes in AAC. In accordance with the breeding regions, the *Wx^a^*/*Wx^b^* ratio of UYR was lower than those of SC and MLYR, consistent with AAC, indicating that *Wx* is the main gene regulated by AAC synthesis.

*ALK/SSIIA* encodes soluble starch synthase IIA (SSIIA) in rice, which plays a specific role in the synthesis of long B1 chains by elongating the short A and B1 chains of amylopectin in the endosperm [15,32,34,44]. SSIIa is the key enzyme controlling GT in rice [45]. Several studies have reported that at least four SNPs of *ALK* are associated with the four alleles in rice (*ALK^a^*, *ALK^b^*, *ALK^c^* and *ALK^d^*) [15,32,33,46,47]. Only two allele types, *ALK^b^* and *ALK^c^*, were detected in the experimental material tested herein. Previous results show that GT was higher for *ALK^c^* and lower for *ALK^b^*, consistent with our current results. *ALK^a^* was generally present in *GJ*, and all rice accessions selected were *XI*, consistent with previous reports [34,48,49]. Over the years, the *ALK^c^*/*ALK^b^* ratio of RLs fluctuated irregularly, and based on area, UYR was significantly higher than others, consistent with the GT data. This indicates that *ALK* is the main gene regulating GT in rice. The *ALK^c^*/*ALK^b^* ratio in IRVs was lower than in 2-RLs and 3-RLs, consistent with *T_p_* values reported in previous work.

The genetic material affecting rice ECQ in hybrid rice comes from RLs and sterile lines, and the two main genes (*Wx* and *ALK*) regulating rice ECQ are separated in F_2_ plants, resulting in heterozygous rice eaten by consumers [50]. In theory, the endosperm of hybrid combinations includes four genotypes, *Wx^b^*/*Wx^b^*/*Wx^b^*, *Wx^b^*/*Wx^b^*/*Wx^a^*, *Wx^b^*/*Wx^a^*/*Wx^a^* and *Wx^a^*/*Wx^a^*/*Wx^a^*, and the ratio of genetic separation of these four genotypes is 1:1:1:1 [51]. The AC phenotype in the heterozygous state containing *Wx^a^* tends to be *Wx^a^*/*Wx^a^*/*Wx^a^*. If the sterile line is the *Wx^a^* type, even if the RL is *Wx^b^*, 3/4 of the final edible hybrid rice will be high in AC, resulting in hybrid rice with a lower ECQ. *Wx* alleles and AC of sterile lines have been assessed alongside the main mating combinations II-32 A, Jin23 A, Zhenshan97 A, Zhong9 A, Bo A, Long tefu A, Y58A, Xieqing Zao A, Peiai 64S and Gang 46A, the widely used areas Zhenshan 97A, Wei 20A, II-32A, Jin 23A, Gang 46A, Xieqing Zao A, Bo A, Long Tefu A, Peiai 64S and Zhong 9A and corresponding maintainer line materials (https://www.ricedata.cn/variety/) (accessed 1 January 2022). Based on the statistics for sterile lines of females matching RLs in our experiment, the results are consistent with the above website (Appendix A). Almost all sterile lines were *Wx^a^* with a high AC, which may be the main reason for the low ECQ of hybrid rice [17,24]. In addition, *ALK* will also be separated in the F_2_ generation, leading to differences in the degree of gelatinisation of hybrid rice during cooking, which further reduces ECQ [35].

There are two types of *XI* hybrid rice, three-line hybrids and two-line hybrids, and the corresponding RLs are 2-RLs and 3-RLs [11]. Hybrid rice bred from RLs were analysed, and the results show that in the breeding process, we relied heavily on several or even a single backbone RL, and many varieties were bred over a short time period; hence, the assimilation of these varieties was rapid. Hybrid rices and their male parents (RLs) that had been employed in the last 5 years were analysed, and of all RLs, only 80 were bred as male parents. It is worth mentioning that many RLs have been eliminated by breeders. Hybrid rice bred as a parent from several varieties accounts for the majority of hybrids developed over the past 5 years. In particular, Huazhan has been used for 94 varieties, accounting for 29% of the total. In the process of hybrid rice breeding, the high-frequency utilisation of a few backbone RLs will decrease the diversity of rice quality traits. Wushansimiao and Chenghui727 have also been heavily relied upon, although not to the same extent as Huazhan. ECQ analysis of RL materials used with high frequency may reveal information on taste value, and almost all varieties tested herein were *Wx^b^* haplotypes.

We need to consider improving the yield and broad-spectrum resistance of hybrid rice in relation to RLs [50]. In addition, we need to pay attention to rice ECQ, which can be enhanced by introducing major genes such as *Wx* alleles [51]. Rice with a low AC is less expansive, fluffy and soft, and hence popular among consumers [52]. Because of its adaptability to almost all climatic situations, many varieties of rice are present worldwide, with differences in ECQ, and a favoured type of rice for one cultural group may not be favoured by others [17]. The rice (*XJ*) variety available in northern and eastern parts of Asia is sticky and soft when cooked. Introducing alleles that regulate the synthesis of AAC and lower it compared with *Wx^b^*, such as *Wx^op^*, *Wx^mp^* and *Wx^mw^/^la^* can improve ECQ [23,29,30]. Similarly, according to the different requirements for populations in different regions for high-quality rice, we should cultivate targeted rice varieties.

Previous studies and our current results show that *ALK* had no significant effect on taste value, and there was no strict requirement for the GT of high-quality rice, resulting in no selection pressure for *ALK*. However, previous studies showed that the taste value of *ALK^c^* rice after cooling was significantly lower than that of *ALK^b^* rice [51]. The gelatinisation temperature increases the cooking time of rice, increasing the need for fuel, which has a great impact on rice-related food processing. Ultimately, diversifying varieties should provide breeders with more raw materials (parents) for selecting and developing desirable traits.

In summary, we compared RLs and conventional materials employed in the rice breeding process over the past 50 years. The overall ECQ of RLs has remained high and not fluctuated significantly. The main reason for the improvement in ECQ in hybrid rice may be derived from sterile lines of female parents. RLs can use other alleles of *Wx*, such as *Wx^op^*, *Wx^mp^* and *Wx^mw^*, to further reduce AC and improve ECQ, which may further improve hybrid rice ECQ. RLs can increase the utilisation of the *ALK^b^* allele and improve the ECQ of hybrid rice varieties. New results should be compared with those of previous studies, and any findings and their implications should be discussed in the broadest possible context.

## 4. Materials and Methods

### 4.1. Plant Materials and Growth Conditions

A total of 387 indica rice accessions provided by the Hunan Hybrid Rice Research Center, Changsha, China, were used in this study (Figure 1a and Appendix A). They included 112 varieties of IRVs, 59 2-RLs and 216 3-RLs, representing the parents of the majority of *XI* hybrid rice accessions that have been widely planted in China over the last 50 years, along with some varieties from foreign sources. All cultivation was performed in the same field in Changsha, China. The planting season was from May to November.

### 4.2. Polished Rice and Flour Preparation

Seeds were de-husked using a rice huller SY88-TH (SSANG YONG Motor, Seoul, Korea) and milled with a grain polisher (Kett, Tokyo, Japan). Intact rice was selected to measure the taste value, and another portion of polished rice samples was ground into flour in a mill (FOSS 1093 Cyclotec Sample Mill; Foss A/S, Hillerød, Denmark) and passed through a 100-mesh sieve. Flour was incubated in a 37 °C oven for 48 h, then in the natural environment for 2 days, and stored at 4 °C. Taste value data included taste value, hardness and stickiness.

### 4.3. Determination of Grain Physicochemical Properties

AAC of flour was measured by iodine colorimetry. [53] with modifications. GT of flour was determined by DSC (DSC200F3, NETZSCH, Bavaria, Germany) [16]. DSC data included enthalpy of gelatinisation (Δ*H*), onset temperature of gelatinisation (*T_o_*), peak temperature of gelatinisation (*T_p_*) and terminating temperature of gelatinisation (*T_e_*). Pasting properties were determined by RVA (Newport Scientific, Warriewood, Australia) [54]. RVA data included hot-paste viscosity (HPV), peak paste viscosity (PKV), breakdown value (BDV), cool-paste viscosity (CPV) and setback value (SBV). All tests were performed in triplicate.

### 4.4. Measurement of Rice Taste Value

A STA1B rice taste meter (Sasaki, Tokyo, Japan) was used to determine the taste values of all materials. ECQ mainly reflects the taste value, hardness and stickiness of rice. All tests were performed in triplicate.

### 4.5. Sequencing and Genotype Annotation

DNA sequencing and SNP calling for all rice accessions were performed as described previously [11]. Tag SNPs were then identified from each clump using De Bakker’s algorithm implemented in Haploview [55]. Linkage disequilibrium (LD) blocking these tag SNPs was then assessed using Gabriel’s algorithm [56] and visualised using Haploview [42]. SNPs within a clump were selected based on *p* < 0.01 and an LD value of *r*^2^ > 0.5 with index SNPs.

### 4.6. Detection of Wx and ALK Alleles

Genomic DNA was extracted from fresh leaves of all parental lines using a modified CTAB method. *Wx* and *ALK* allelic variations were analyzed by KASP genotyping [38]. The details of allele-specific primers are shown in Appendix A.

### 4.7. Statistical Analysis

DNA sequences of *ALK* genes of different materials were obtained, and homologous alignment was performed using the NCBI website (http://www.ncbi.nlm.nih.gov/) (accessed 1 January 2022). Nucleotide sequence similarity and multiple sequence alignment analyses were carried out using ClustalX and GeneDoc. A phylogenetic tree was constructed by the neighbour-joining (NJ) method with the Kimura 2-parameter (K2-P) model using MEGA version 5.0. Experiments were carried out in triplicate, and results are reported as mean values ± standard deviations (SD). One-way analysis of variance (ANOVA) and Tukey’s multiple comparison test were used to determine significant differences among mean values using SPSS 16.0 statistical software (IBM, Armonk, NY, USA).

## Figures and Tables

**Figure 1 ijms-23-05941-f001:**
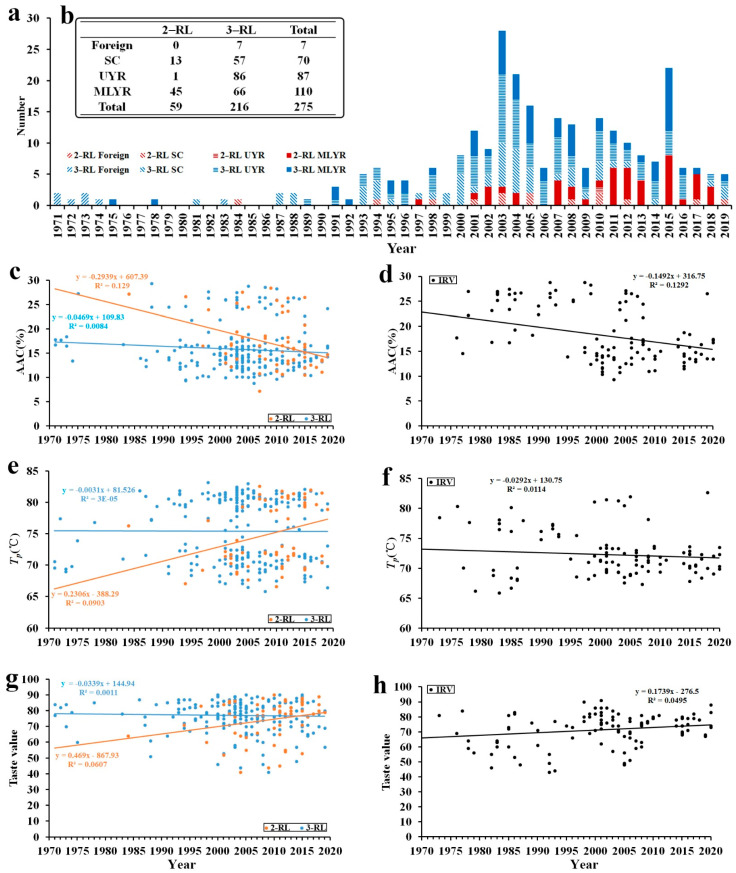
Phenotypic distribution statistics over time. (**a**,**b**) Distributions of materials in terms of breeding time, region and type. (**c**,**e**,**g**) 2-RLs and 3-RLs. (**d**,**f**,**h**) IRVs. (**c**,**d**) Apparent amylose content. (**e**,**f**) Peak temperature of gelatinisation. (**g**,**h**) Taste value.

**Figure 2 ijms-23-05941-f002:**
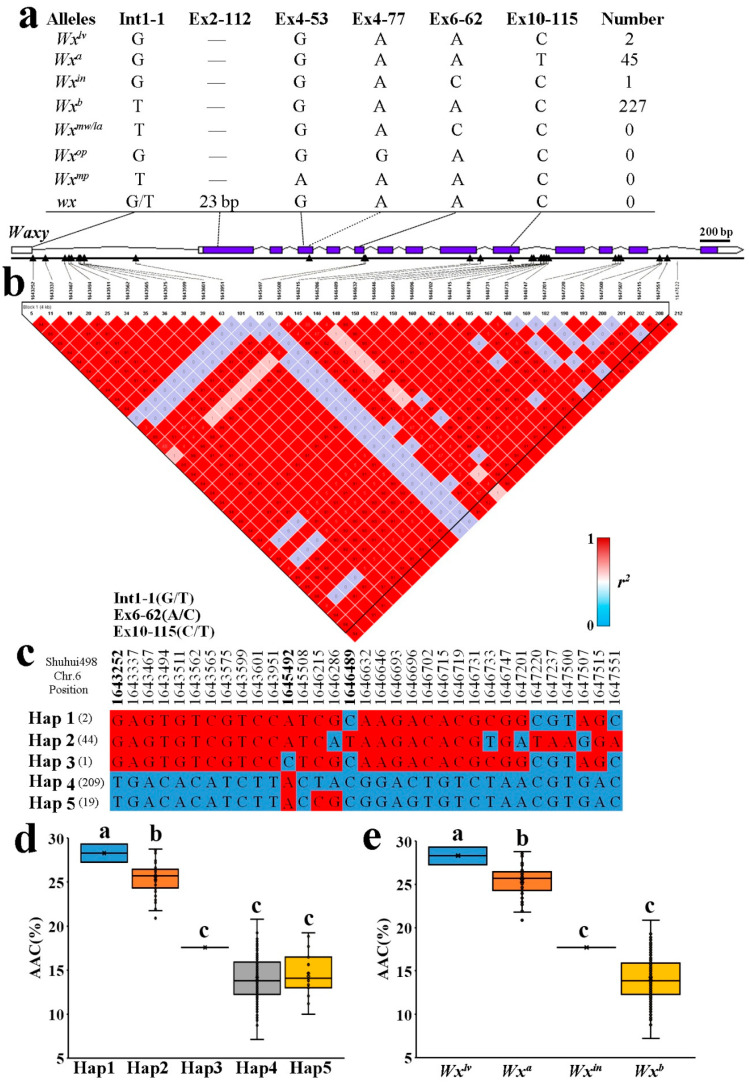
Analysis of different *Wx* alleles and haplotypes. (**a**) *Wx* allele types and numbers screened by KASP marker. (**b**) Linkage disequilibrium (LD) analysis. The *r*^2^ value is reflected on the matrix diagram. (**c**) Results of haplotype analysis. (**d**) Apparent amylose content of different haplotypes. (**e**) Apparent amylose content of different allele types. Different letters indicate significant differences (*p* < 0.05).

**Figure 3 ijms-23-05941-f003:**
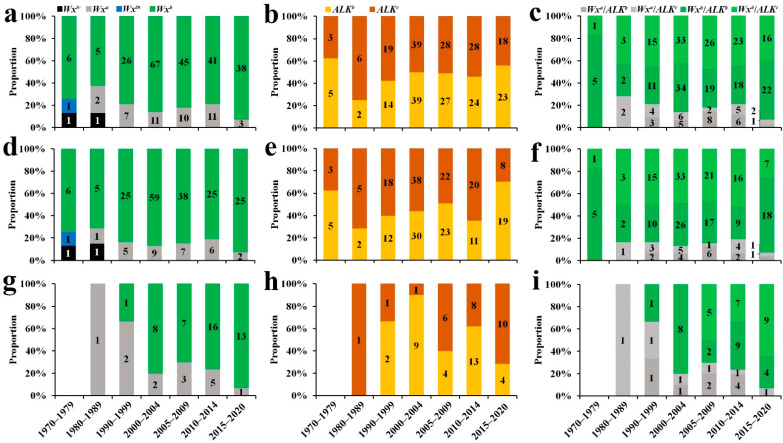
Frequency distribution of different rice materials based on different alleles. (**a**,**d**,**g**) Frequency distribution of different alleles of *Wx*. (**b**,**e**,**h**) Frequency distribution of different alleles of *ALK*. (**a**–**c**) Total restorer lines. (**d**–**f**) 3-RLs. (**g**–**i**) 2-RLs.

**Figure 4 ijms-23-05941-f004:**
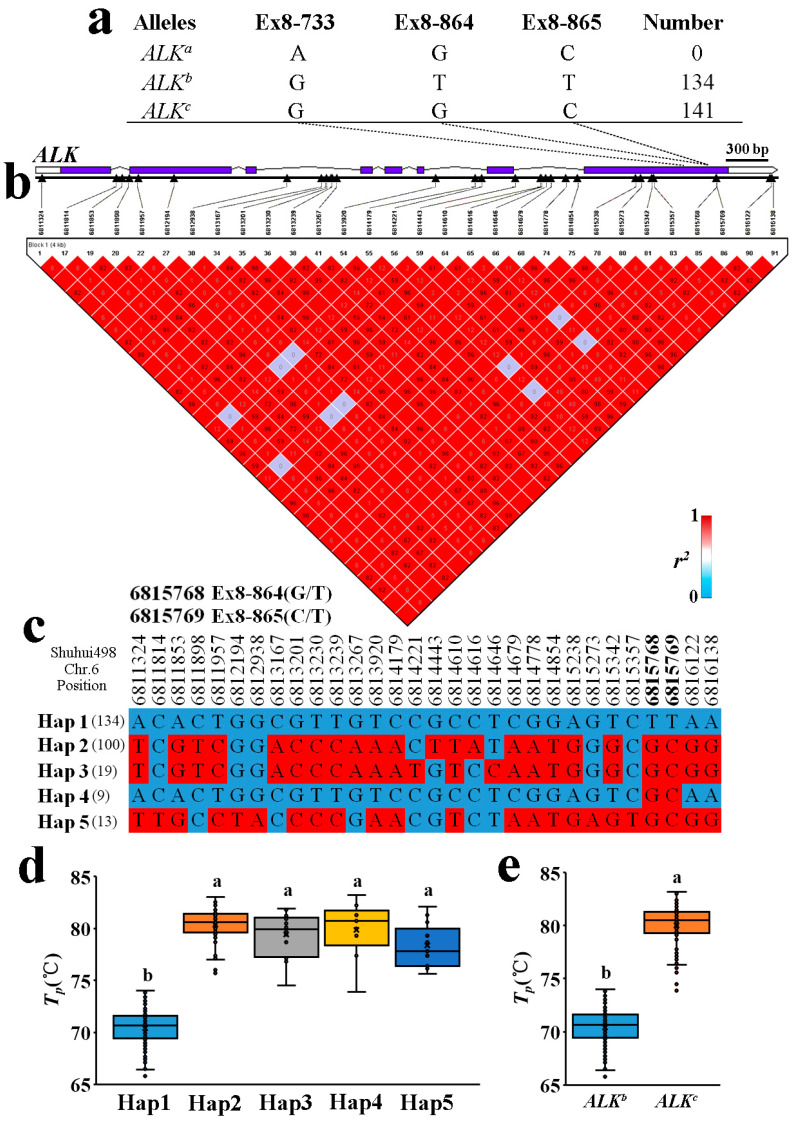
Analysis of different alleles and haplotypes of *ALK*. (**a**) Allele types and number of *ALK*s screened by KASP markers. (**b**) Linkage disequilibrium (LD) analysis. The *r*^2^ value is shown on the matrix diagram. (**c**) Haplotype analysis. (**d**) Peak temperature of gelatinisation of different haplotypes. (**e**) Peak temperature of gelatinisation of different allele types. Different letters indicate significant differences (*p* < 0.05).

**Figure 5 ijms-23-05941-f005:**
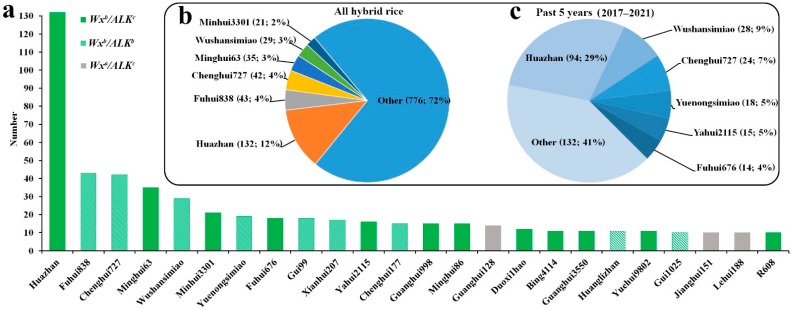
Hybrid rice varieties bred using RLs. (**a**) RLs of male plants used to breed > 10 hybrid rice varieties. (**b**) All hybrid varieties cultivated using the restorer line in this experiment. (**c**) Hybrid varieties cultivated in the past 5 years using the restorer line in this experiment.

## Data Availability

Datasets supporting the conclusions of this article are included within the article (and its additional files).

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
