# Peer review of "Allelic Diversification of the Wx and ALK Loci in Indica Restorer Lines and Their Utilisation in Hybrid Rice Breeding in China over the Last 50 Years"

_ijms, 2022, doi:10.3390/ijms23115941_

Round 1

Reviewer 1 Report

Eating and cooking quality (ECQ) is one of the important objectives for rice breeding, and starch properties significantly affect ECQ of rice. Among them, amylose content and amylopectin branch chain length are major factors, which is controlled by Wx and SSIIa (ALK) among rice varieties. The present study focused on the alteration of the ECQ-related phenotypes and haplotypes at Wx and ALK among Chinese hybrid rice varieties over 50 years, and found noticeable trend in recent years. The results seem to include significance for readers, but still need several revisions for the reasons below.

Major points
1) Figure 1, The authors focused on RLs, but is there any reason for paying no attention to GMS lines? In case previous studies already analyzed GMS lines, it should be mentioned in the introduction.

2) Figure 1c-h, it seems to be difficult to understand and compare the general alteration of AAC, Tp, and Taste value between RL and IRL from the dot plot. Adding line graph or statistical analyses will be helpful for readers. Figure 1c-h is the foundation of the present study, thus the authors should make some more effort to compare statistically.

3) p6, line175, if the authors consider "the reason for the decrease in AAC for IRVs before the 2000s is that the allelic proportion of the Wxb type increased", the relationship between AAC and Wx haplotypes should be shown. They can easily show by changing the color of Figure 1d by the Wx haplotypes.

4) p6, line177-183, I cannot understand on which results the authors describe these sentence, because the label of Figure 4 is inappropriate. The figures exist from a-i, but figure legends from a-f.

5) p8, line233, the location of Waxy must be wrong.

6) p8, line245-247, refer Figure 4 appropriately. And need fixation of labels in Figure 4.

Minor points
1) Abbreviation should be appropriately summarized in the Abbreviation part. They didn't cover all of the abbreviation used in the manuscript.

2) DBV in TableS2 or p3 may be BDV.

3) p3, line116, Table S2 should be appropriately referenced after "r = −0.515, p <0.01" like line119.

4) For the easier understanding of readers, it is desirable to explain the abbreviation used in Figure 1 in the legends such as SC and UYR.

5) The graphic pattern of 3-RL Foreign and 3-RL SC is difficult to distinguish.

6) Many of "Wx" and "ALK" should be italicized in the sentence.

7) Figure S7 and S9b, materials should be described in the title or legends. In case it contains IRV, 2-RL and 3-RL, it should be described so.

8) Figure 4c,f,i, the gray colors are difficult to distinguish.

9) Figure 5a, the title of Y axis is not informative. In addition, the title of Figure 5b,c is not appropriate. The authors need to include enough information into the title or legends other than Figure 5, too.

Author Response

Thank you very much for your positive comments. We have revised it according to your comments. Please see the attachment.

Reviewer 2 Report

In this study, authors investigated the allelic diversification Waxy (Wx) and alkaline denaturation (ALK) loci in rice and their utilization in hybrid rice breeding in China over the last 50 years. In my opinion, the study is planned nicely, and the findings are interesting. The data looks robust, and manuscript is written well. However, I have few comments that can be considered for improving the quality of manuscript.

·        I would suggest authors to modify the title to make it more concise and appealing for a wider readership.

·        Line 30: Analysis of hybrid rice varieties and RLs in the last “50” years. There are many more typographical errors in the manuscript. Authors need to check them carefully.

·        Please make a separate section for conclusion.

·        It would be good to include future prospects of the study in the conclusion section.

·        There are many places where the author's message is not clear due to improper phrasing and English language. Authors need to check that throughout the manuscript.

Author Response

(The authors gave the same response as above.)
